Clustering of fMRI data: the elusive optimal number of clusters

Seghier Mohamed L. mseghier@gmail.com
Cognitive Neuroimaging Unit, Emirates College for Advanced Education , Abu Dhabi , United Arab Emirates
Abdullah Jafri
Electronic publication date: 2018 Oct 3
Publication date: 2018
Volume: 6
Electronic Location ID: e5416
Received 2018 May 6; Accepted 2018 Jul 19
Copyright: ©2018 Seghier
Copyright year: 2018
Copyright holder: Seghier
License: This is an open access article distributed under the terms of the Creative Commons Attribution License, which permits unrestricted use, distribution, reproduction and adaptation in any medium and for any purpose provided that it is properly attributed. For attribution, the original author(s), title, publication source (PeerJ) and either DOI or URL of the article must be cited.
License URL: https://creativecommons.org/licenses/by/4.0/

Keywords: Functional MRI, Data-driven analysis, Unsupervised fuzzy clustering, Brain networks, Cluster validity, Fuzzy compactness and separation

Funding: ECAE’s Research Office This work was supported by the ECAE’s Research Office. The funders had no role in study design, data collection and analysis, decision to publish, or preparation of the manuscript.

==============================
Model-free methods are widely used for the processing of brain fMRI data collected under natural stimulations, sleep, or rest. Among them is the popular fuzzy c-mean algorithm, commonly combined with cluster validity (CV) indices to identify the ‘true’ number of clusters (components), in an unsupervised way. CV indices may however reveal different optimal c-partitions for the same fMRI data, and their effectiveness can be hindered by the high data dimensionality, the limited signal-to-noise ratio, the small proportion of relevant voxels, and the presence of artefacts or outliers. Here, the author investigated the behaviour of seven robust CV indices. A new CV index that incorporates both compactness and separation measures is also introduced. Using both artificial and real fMRI data, the findings highlight the importance of looking at the behavior of different compactness and separation measures, defined here as building blocks of CV indices, to depict a full description of the data structure, in particular when no agreement is found between CV indices. Overall, for fMRI, it makes sense to relax the assumption that only one unique c-partition exists, and appreciate that different c-partitions (with different optimal numbers of clusters) can be useful explanations of the data, given the hierarchical organization of many brain networks.

Introduction

There are many contexts where model-based methods are inadequate to map brain function, including for instance tasks that cannot be fully controlled (e.g., sleep, learning, natural stimulation, continuous rest; Bartels & Zeki, 2004; Bartels & Zeki, 2005; Hasson et al., 2004; Lee et al., 2012; Malinen, Hlushchuk & Hari, 2007; Zacks et al., 2001) or when the hemodynamic correlates of neural activity are altered in unknown ways (e.g., patients with impaired vasculature). In such cases, approaches without a priori knowledge, known also as model-free or data-driven methods, are of great help.

Several data-driven methods have previously been used in fMRI (DonGiovanni & Vaina, 2016; Thirion et al., 2014), including fuzzy clustering (Baumgartner, Windischberger & Moser, 1998; Fadili et al., 2000; Golay et al., 1998; Jahanian et al., 2004) and independent component analysis (McKeown et al., 1998). These methods have been used in many scenarios to extract meaningful information from fMRI data in the absence of any prior knowledge (Aljobouri et al., 2018; Baumgartner et al., 2000; Lange et al., 2004; Ma et al., 2011; Smolders et al., 2007; Tang et al., 2015; Wismuller et al., 2004). One popular data-driven clustering method is based on the classic fuzzy c-mean (FCM) algorithm (Bezdek, 1981). Although FCM allows high computational flexibility, its robustness may depend on several methodological issues. Specifically, these include the initialisation problem, the choice of similarity or distance metric, and the usually unknown optimal number of classes or prototypes (e.g., Alexiuk & Pizzi, 2004; Esposito et al., 2002; Fatemizadeh, Taalimi & Davoudi, 2009; Jahanian, Soltanian-Zadeh & Hossein-Zadeh, 2005; Lange et al., 2004; Moller et al., 2002; Quiqley et al., 2002; Soltanian-Zadeh et al., 2004; Windischberger et al., 2003). This study focuses on the issue of the optimal number of clusters that can be extracted from fMRI data.

It is critical for any reliable clustering method to be able to determine whether: (i) the data contains any structure and (ii) the segregated clusters are ‘true’ representations of the data (Dubes, 1987; Windham, 1981). This issue is generally expressed in terms of the ability of the algorithm, here FCM, to cluster the data into an optimal number of clusters (copt). To do that, previous studies have introduced many measures, called cluster validity (CV) indices, to estimate copt in an unsupervised manner (for a review see Bezdek & Pal, 1998; Hammah & Curran, 2000; Kim & Ramakrishna, 2005; Maulik & Bandyopadhyay, 2002; Wang & Zhang, 2007; Zhou et al., 2014). The rationale behind these CV indices is that a good and useful clustering should yield compact and well-separated clusters. Indeed, it is not surprising that many proposed CV indices combine different measures of compactness (cohesiveness) and separation (isolation) among clusters, and would reach their optimal values for the best c-partition (i.e., data clustered into copt clusters).

A few studies have previously investigated the effectiveness of CV indices in the context of fMRI data clustering (e.g., Alexiuk & Pizzi, 2004; Fadili et al., 2000; Fadili et al., 2001; Goutte et al., 1999; Moller et al., 2002; Seghier & Price, 2009). Some known features of fMRI data may make the clustering particularly challenging (Thirion et al., 2014), including for instance the huge number of points (i.e., voxels) in a typical fMRI dataset, the poor signal-to-noise ratio in fMRI (noisy data), the small proportion of voxels of interest that might be considered as relevant (i.e., an ill-balanced problem), and the presence of artefacts or outliers (i.e., caused by head motion or signal loss). Given this complexity, it might be the case that reliance on a single CV index might not be enough, in particular when the data are noisy and the expected number of clusters is relatively high. Here, the author compared the identified optimal c-partition when applying different CV indices to the same datasets. In particular, the author investigated the behaviour of different measures of compactness and separation when using previously published CV indices. The current study also aims to introduce a new CV index that specifically incorporates suitable compactness and separation measures that are useful for data with larger optimal number of clusters.

Methods

Fuzzy clustering

Our clustering method was based on the popular fuzzy c-mean (FCM) algorithm (Bezdek, 1981; Bezdek et al., 1997). In the context of fMRI, the FCM algorithm can segregate or cluster n brain voxels (feature vectors) into c expected clusters (c ≥ 2). Each voxel i is a vector Xi of p properties (e.g., number of collected volumes or scans). Each cluster j is characterised by a centroid Vj, that represents its characteristic timecourse (prototype). The resemblance between each voxel i and each centroid Vj is assessed by the distance Dij between Xi and Vj. The degree of membership Uij is calculated for each voxel i by comparing Dij for each cluster j to all other clusters.

In brief, the standard FCM algorithm iteratively minimises the following objective function Jm: (1) Jm= ∑i=1n ∑j=1cUijm⋅Dij2

where “m” is the degree of fuzziness.

Degrees of membership U and centroids V are updated as following:

(2) Uij=1∑k=1cDijDik2∕m−1

(3) Vj=∑i=1nUijm⋅Xi ∑i=1nUijm.

Optimal clustering depends on the choice of the similarity D, the degree of fuzziness m and the optimal number of clusters copt, as detailed below.

Similarity measure D

Here I used a modified version of the hyperbolic correlation distance proposed previously by Golay et al. (1998). In their work, D was defined as (Golay et al., 1998): (4) Dij=1−CCij1+CCij.

Where CCij is the Pearson correlation coefficient between Xi and Vj.

Here, a modified version of D was used: (5) Dij=CCij−CCijCCij+CCij.

This new formula uses the square root function, a monotonically increasing function over x > 0 that satisfies the following inequality: x≥x, for x ∈ [0, 1]. The rationale here was to increase the difference (i.e., discrimination power) between relatively close correlation values in particular between mid and high correlations (cf. Fig. S1).

Optimal number of clusters

A good and robust clustering should yield compact and well-separated clusters. This is assumed to be the case when the number of clusters reaches an optimal value copt. The exact copt value is however unknown in fMRI data. Previous reports have suggested that copt can be found within the interval [2, n] (Zahid, Limouri & Essaid, 1999); however the exact copt can only be estimated empirically. Typically, FCM is repeated several times with different c values (i.e., equivalent to an unsupervised fuzzy clustering analysis Fadili et al., 2001) and the c value that optimises a given criterion, here a given CV index, is considered as the optimal copt, and that criterion is typically defined as a trade-off between compactness and separation.

Before introducing the different CV indices used here, it might be helpful to define the core measures of compactness and separation using unified mathematical notations. These measures can be seen as building blocks that can be combined into different CV indices. Ultimately, the definition of those measures would help appreciate the inherent links (or similarity) between previously suggested CV indices, before introducing the rationale of the new CV index.

Compactness and separation measures

Two core quantities, noted nm,j and σm,j, were defined as following:

(6) nm,j= ∑i=1nUijm

(7) σm,j= ∑i=1nUijm⋅Dij2.

The measures n1,j and n2,j represent the fuzzy cardinality and the fuzzy partition of cluster j respectively. The quantity σm,j denotes the fuzzy variation of cluster j, though other studies have instead used σ1,j as a measure of fuzzy variation (e.g., Gath & Geva, 1989; Rezaee, Lelieveldt & Reider, 1998; Sun, Wang & Jiang, 2004).

Those core quantities can then be combined into different forms to give away different measures of fuzzy compactness (cohesiveness) for a given c-partition. Using similar notation as previous studies, quantities called πm,1 (Bensaid et al., 1996; Zahid et al., 1999), πm,m (Bouguessa, Wang & Sun, 2006), and FC (Fadili et al., 2001; Zahid et al., 1999) were computed as following:

(8) πm,1= ∑j=1cσm,jn1,j

(9) πm,m= ∑j=1cσm,jnm,j

(10) FC=∑i=1nmaxjUij2 ∑i=1n maxjUij.

Likewise, the fuzzy separation (isolation) between clusters was previously estimated with several fuzzy separations quantities called Km (Fukuyama & Sugeno, 1989), FS (Fadili et al., 2001; Zahid et al., 1999), S (Zahid et al., 1999) and SS (Rezaee, Lelieveldt & Reider, 1998):

(11) Km= ∑j=1cnm,j⋅Vj−X¯2

(12) FS= ∑j=1c−1 ∑k=1c−j ∑i=1nminUij,Ui,k+j2 ∑i=1n minUij,Ui,k+j

(13) S=1c∑j=1cVj−X¯2

(14) SS= ∑j=1c1∑k=1cVj−Vk

where X¯ stands for the global mean of the whole data.

Interestingly, the ratio FS/FC (i.e., separation divided by compactness) is known as the fuzzy overlap (FO) coefficient (see Fadili et al., 2000 for more details).

Furthermore, different measures of between-centroid distance have been proposed, including the minimum distance Vdmin (e.g., Schwämmle & Jensen, 2010; Xie & Beni, 1991), the maximum distance Vdmax (e.g., Rezaee, Lelieveldt & Reider, 1998), and the minimum distance Vdmin,j between a cluster j and the remaining clusters (Wu & Yang, 2005):

(15) Vd min= minj,kVj−Vk

(16) Vd max= maxj,kVj−Vk

(17) Vd min,j= mink≠jVj−Vk.

These measures, based on the distance between estimated centroids, can be seen as alternative separation measures. They can be handy when the clustering is showing redundant clusters.

This section introduces two new measures of separation and discrimination between voxels by combining different measures of fuzzy cardinality and variation (cf. Eqs. (6) and (7)): a fuzzy intra-cluster (IDintra) dissimilarity coefficient and an inter-cluster (IDinter) dissimilarity coefficient:

(18) IDintra= maxjn−n1,jn1,j⋅σ1,j ∑k=1,k≠jcσ1,k

(19) IDinter= minj mink,k≠jσ1,kσ1,j.

Small IDintra values would indicate that, across all clusters, voxels that are close to a given cluster are well-isolated from voxels that are far from that cluster, whereas high IDinter values indicate well-discriminated voxels (i.e., small fuzzy overlap between clusters). Our initial tests with noisy simulated data showed the need to define new separation measures that are robust to noise and can handle c-partitions with higher number of clusters, hence the new definitions in Eqs. (18) and (19).

Cluster validity measures

There are many CV indices in the literature (probably more than 50 indices), hence it is beyond the scope of this study to test all of them. In a preliminary analysis (results not shown here), about 20 selected CV indices were first tested on several simulated datasets (as defined in Bezdek & Pal, 1998; Bouguessa, Wang & Sun, 2006; Dave, 1996; Fukuyama & Sugeno, 1989; Geva et al., 2000; Kim, Park & Park, 2001; Kim, Lee & Lee, 2003; Kim & Ramakrishna, 2005; Kwon, 1998; Pakhira, Bandyopadhyay & Maulik, 2004; Pakhira, Bandyopadhyay & Maulik, 2005; Pal & Bezdek, 1995; Rezaee, Lelieveldt & Reider, 1998; Rhee & Oh, 1996; Sun, Wang & Jiang, 2004; Tsekouras & Sarimveis, 2004; Wu & Yang, 2005; Xie & Beni, 1991; Yu & Li, 2006; Zahid et al., 1999; Zahid, Limouri & Essaid, 1999). These CV indices were selected from earlier studies (for a similar rationale, see recent comparison study Zhou et al., 2014), and many of them are well-established indices. Some of these CV indices have been used in previous fMRI studies. More recent CV indices (e.g., see He, Tan & Fujimoto, 2016; Hu et al., 2011; Lin et al., 2016; Ren et al., 2016; Rezaee, 2010; Yang et al., 2018; Zhang et al., 2014) were not explicitly tested here.

From this preliminary analysis, seven CV indices (out of twenty) were selected according to the following four criteria: CV indices should (i) combine both measures of separation and compactness; (ii) not suffer from monotonic dependency with the number of expected clusters; (iii) not necessitate the categorisation or the binarisation of U (i.e., crisp degrees of membership) during CV computation; (iv) be fast to compute when n is expected to be very high (e.g., hundreds of thousands of voxels in the context of fMRI data). The seven selected CV indices that satisfied the different criteria are described below and listed in Table 1.

(1)- The Rezaee-Lelieveldt-Reider index CVRLR (Rezaee, Lelieveldt & Reider, 1998): (20) CVRLR=∑j=1cσ1,jc⋅σX+1α⋅Vd max⋅SSVd min.

The constant α is a weighting constant and σX is the variance of the whole data set. The best c-partition is obtained by minimising CVRLR with respect to the number of clusters c. In the original definition of CVRLR, the constant α was set to 1; however, here α was set to the value of Vd maxVd min⋅SS at the maximum number of clusters (cmax) as suggested previously (Sun, Wang & Jiang, 2004).

Table 1 List of the selected cluster validity (CV) indices.

CV index	Proposed by	Range	Value at copt	
CVRLR	Rezaee-Lelieveldt-Reider index (1998). A modified version was used here (Sun, Wang & Jiang, 2004)	[0, +∞[	Minimal	
CVZLE	Zahid-Limouri-Essaid index (1999)	]−∞, +∞[	Maximal	
CVGV	Geva index (2000)	[0, +∞[	Maximal	
CVKP	Kim-Park index (2001)	[0, +∞[	Minimal	
CVPBM	Pakhira-Bandyopadhyay-Maulik index (2004)	[0, +∞[	Maximal	
CVWY	Wu-Yang index (2005)	[ −c, c]	Maximal	
CVBWS	Bouguessa-Wang-Sun index (2006)	[0, +∞[	Maximal	
CVnew	A new CV index	[0, +∞[	Maximal	

(2)- The Zahid-Limouri-Essaid index CVZLE (Zahid et al., 1999; Zahid, Limouri & Essaid, 1999): (21) CVZLE=α⋅Sπm,1−FSFC.

The constant α is independent from c and was introduced here as a scaling factor to take into account the difference in values between the two subtracted quantities. The constant α was set here to the value of the fuzzy overlap (FS/ FC) at c = cmax (note that in the original paper of Zahid et al., α was set equal to 1). The best c-partition is obtained by maximising CVZLE with respect to c. This CVZLE index has previously been used for fMRI analysis (Fadili et al., 2001).

Note that the ratio Sπm,1 in Eq. (21) is also known as the Pal-Bezdek cluster validity index (Pal & Bezdek, 1995).

(3)- Among several CV indices suggested by Geva and colleagues (Geva et al., 2000), the invariant index CVGV was selected here to measure the ratio of the between-cluster scatter matrix to the within-cluster scatter matrix (Geva et al., 2000): (22) CVGV=K1c2⋅J1.

The normalisation with the number of clusters c minimise the monotonically increase of CVGV when c increased. This index should be maximal at the optimal c-partition.

(4)- The Kim-Park index, noted CVKP (Kim, Park & Park, 2001): (23) CVKP=π1,1c+1α⋅cVd min.

The best c-partition is obtained by minimising the index CVKP with respect to the number of clusters c. This index has previously been used for fMRI analysis (Moller et al., 2002).

(5)- The Pakhira-Bandyopadhyay-Maulik index CVPBM (Pakhira, Bandyopadhyay & Maulik, 2004; Pakhira, Bandyopadhyay & Maulik, 2005): (24) CVPBM=αc⋅Vd maxJm.

With α as a constant term (e.g., α was set here to n). The best c-partition is obtained by maximising CVPBM with respect to the number of clusters c.

(6)- The Wu-Yang index CVWY (Wu & Yang, 2005): (25) CVWY= ∑j=1cn1,j maxjn1,j−exp−Vd min,j2S.

This index compared the fuzzy partition of each cluster to its exponential separation, with −c < CVWY < c, and CVWY is maximal at copt.

(7)- The Bouguessa-Wang-Sun index CVBWS index (Bouguessa, Wang & Sun, 2006): (26) CVBWS=Kmπm,m.

This index CVBWS should be maximised with respect to c.

(8)- Our new CV index, noted CVnew, combined different measures of compactness and separation as following: (27) CVnew=Km⋅IDinterIDintra⋅FCJ1.

The best c-partition should maximise CVnew. The rationale behind incorporating those specific compactness and separation measures (IDinter, Km, FC, IDintra, J1) in the definition of CVnew is illustrated below with simulated (noisy) datasets.

Simulated data

Twenty-two simulated datasets were generated as following. First, a fixed number c of time-courses with p datapoints (p = 100) were generated from a unit normal distribution (mean = 0, σ = 1). The Pearson correlation between these c time-courses was less than 0.1 for all simulated datasets. Second, each time-course was replicated rj times, with j = 1…copt, and ∑j=1crj=n (where n is the total number of voxels, set here to 1,000). Third, n random timecourses with p datapoints, generated from a normal distribution (mean = 0) but with variable noise levels (σ = 1 or 4) were added to the replicated time-courses. This would help to test the robustness of FCM at different noise levels (for a similar rational see Kim, Park & Park, 2001; Wang & Zhang, 2007) and to monitor the behaviour of the different CV indices when the fuzzy compactness of clusters became very low (i.e., high intra-class dissimilarity in noisy data). This procedure generated a dataset X of n voxels, each with p datapoints, with known and fixed numbers of classes. Multidimensional scaling (MDS) tools were used to visualise the simulated c clusters.

Specifically, the following 22 datasets were generated: (i) a single-cluster dataset (noted 1-cluster; e.g., the ‘null’ case, see Tibshirani, Walther & Hastie, 2001) with highly similar voxels (copt = 1; Fig. S2A); (ii) a dataset without any obvious structure (noted n-cluster data; copt near to n; Fig. S2B), see Suleman (2017); (iii) ten datasets with known number of clusters copt varying from 2 to 11 and low noise level (σ = 1, see illustration in Fig. S2C with copt = 3); (iv) ten datasets with a known number of clusters copt varying from 2 to 11 and high noise level (σ = 4, see illustration in Fig. S2D with copt = 3).

All simulated datasets were clustered by FCM with c (i.e., number of expected clusters) varying between cmin = 2 and cmax = 19. All analyses were carried out with homemade Matlab-based scripts (MathWorks, Natick, MA, USA).

Real fMRI data

Real data consisted of single subject fMRI data with a block paradigm design (freely available at: http://www.fil.ion.ucl.ac.uk/spm/data/auditory.html). The block paradigm consisted of alternated epochs between rest and auditory stimulation. 96 volumes were acquired on a modified 2T Siemens MAGNETOM Vision system (TR = 7s, 64 contiguous slices). To avoid T1 effects in the initial scans, the first 12 scans were discarded, leaving 84 scans for further analysis (p = 84). The data were realigned, normalised (voxel size 2 × 2 × 2) and smoothed (FWHM = 6 × 6 × 6 mm). This dataset was selected because it has been used in many previous studies with clustering techniques including FCM (e.g., Gu et al., 2005; Lu, Jiang & Zang, 2004). FCM was applied on this real fMRI dataset with c varying between cmin = 2 and cmax = 39. To identify relevant FCM cluster(s) with activated auditory regions, the centroids (prototype) Vj (j = 1…c) were correlated with the experimental block design (Bandettini et al., 1993).

To appreciate the distribution of brain regions’ sizes in each c-partition, a morphological granulometry was applied to all identified clusters after binarization (Soille, 2003). This analysis estimated the size of each spatially distinct region or blob (26-connected neighbourhood) for a given crisp FCM partition. Given that each voxel belongs to all clusters at different degrees of membership (cf. Uij in Eq. (2)), the threshold was set to 0.5 so that each voxel belongs maximally to one cluster. Practically, for a given c-partition and for each binary cluster j (j = 1…c), the size of each region as well as the number of isolated voxels (i.e., single-voxel regions) were calculated.

This dataset was also analysed with SPM12 software package (Wellcome Trust Centre for Neuroimaging, London UK; http://www.fil.ion.ucl.ac.uk/spm/) using standard procedures. This allowed auditory activations to be identified using model-based methods.

Degree of fuzziness m

The degree of fuzziness m might influence the output of clustering (e.g., Bezdek, 1981; Fadili et al., 2000; Fadili et al., 2001; Krishnapuram & Keller, 1993; Selim & Ismail, 1986; Yu, Cheng & Huang, 2004): when m tends to 1 the classification becomes crisp and Uij takes the value 0 (voxel i is not a member of cluster j) or 1 (voxel i belongs to cluster j) but when m tends to +∞ the classification is purely fuzzy (Uij is near to 1/c). The optimal value of m may depend on the characteristics of the data. Previous empirical work approximated m by a nonlinear function of the dimensions of the data (n and p); for example, Eq. (5) of Schwämmle & Jensen (2010, Page 2845) yields m values of 1.044 and 1.019 for our artificial and real datasets respectively. However, these estimated values are too low compared to typical m values encountered in neuroimaging studies. Previous studies have explored the influence of m on the computation of CV indices (e.g., Zhou, Fu & Yang, 2014), and they found better clustering results with m between 1.2 and 2.5 for fMRI data (Fadili et al., 2000; Fadili et al., 2001; Moller et al., 2002; Smolders et al., 2007).

More specifically, there are two issues to be considered when selecting m during the computation of CV indices. First, several CV indices became inadequate with hard c-partitions (i.e., m tends to 1). Specifically, any measures that are based exclusively on the distribution of U values (e.g., FC, FS) would artificially reach their optimal values independently from the number of clusters c. Second, according to Eq. (3), centroids become close to the mean of the whole data set X¯ when m tends towards +∞. In other words, the c clusters would have comparable fuzzy cardinality values (i.e., Eq. (6)) for larger m values, which may be problematic when some clusters are expected to contain a small number of voxels (see illustration in Fig. S3); for more details see (Selim & Ismail, 1986; Tsekouras & Sarimveis, 2004; Yu, Cheng & Huang, 2004). This issue is particularly critical when analysing task-related fMRI data because activated voxels are expected to represent a small fraction of the whole brain.

Here, m was held to 1.5 throughout this study.

Voxel selection and the ill-balanced dataset problem in fMRI

One important issue during the clustering of fMRI datasets is the selection of the relevant n voxels. Because the number of activated voxels is small (i.e., a few percent) compared to the total number of voxels in a typical whole-brain fMRI dataset, previous studies have suggested different approaches to overcome this ‘ill-balanced’ data problem. For instance, FCM can be limited to relevant voxels within the gray matter, in specific anatomical brain regions, or to voxels with some kind of task-related effects (e.g., see Fadili et al., 2000; Goutte et al., 1999; Gu et al., 2005; Lee et al., 2012; Moller et al., 2002; Seghier, Friston & Price, 2007). Voxel selection might be useful for: (i) reducing the high dimensionality of the problem and improving both computational robustness and speed; (ii) minimising the influence of redundant voxels; and (iii) increasing the accuracy of the clustering by focusing mainly on meaningful voxels. However, the author preferred here to include all brain voxels so that the robustness of the different CV indices can be appreciated when noisy voxels (voxels with no effect of interest) and artefacts are present. FCM was thus applied to all voxels of the real fMRI dataset, yielding a total number of voxels n = 227,716.

FCM convergence and the initialisation problem

Depending on the initialisation of the degrees of membership U (Bezdek, 1981), the FCM algorithm may converge to different c-partitions (e.g., local minima). This problem of initialisation may lead to spurious c-partitions (Moller et al., 2002) when using CV indices. One possible solution is to repeat the FCM algorithm on the same dataset with several different random initialisations (e.g., Moller et al., 2002; Pena, Lozano & Larranaga, 1999), with the expectation that it is unlikely that different starting conditions will lead to the same local minima. Accordingly, for each c value, the FCM algorithm was re-run on the real fMRI dataset ten times with random initialisations (for a similar procedure see Chuang et al., 1999).

Figure 1 Illustration of the behaviour of different measures of compactness and separation.

FCM on the one-cluster (A) and the n-cluster (B) dataset. The number of clusters varied between 2 and 19. See full definition of the different measures in the ‘Methods’.

Results

FCM on simulated data

The 1-cluster dataset

Clustering the 1-cluster dataset (copt = 1) showed how compactness and separation measures behave when data cannot be clustered any further. In this context of high redundancy, it is expected to observe: (i) high similar or identical centroids V, (ii) degrees of membership U near to the fuzziest value 1∕c, and (iii) comparable fuzzy cardinality across clusters. As illustrated in Fig. 1A, the fuzzy compactness FC decreased monotonically with c (i.e., FC = 1∕c) whereas fuzzy separation FS increased linearly with c-1 (i.e., FS = (c − 1)∕2), suggesting a high fuzzy overlap FO between clusters. Likewise, as expected, the fuzzy compactness πm,1 and separation Km showed monotonic dependency with cm−1 and c1−m respectively, suggesting that the product πm,1⋅Km remained constant (independent from c) when data were classified into pure fuzzy clusters. Interestingly, measures of separation based on centroids V (e.g., Vdmin, Vdmax, S) and distances D (e.g., IDintra and IDinter) were independent from c, suggesting highly similar (i.e., identical) centroids V.

The n-cluster dataset

Clustering the n-cluster dataset (i.e., copt towards n) tested the robustness of the different measures of compactness and separation when data is patternless with high dispersion. Compactness coefficients showed similar behaviour as above when clustering the 1-cluster dataset, except for J1 and SS measures. Interestingly, separation measures based on centroids V and distances D showed more complex dependencies with c (Fig. 1B) as compared to the 1-cluster case (Fig. 1A), in particular when using the two new coefficients IDintra and IDinter.

What emerged from above is that Km, IDintra, IDinter, and J1 behaved differently on 1-cluster and n-clusters datasets, which is highly desirable when clustering fMRI data that have complex structure. These results motivated the rationale of including them in the computation of the new CVnew index (as defined in Eq. (27)).

CV indices on data with known numbers of clusters

The different measures of compactness and separation are shown in Fig. 2A for the 7-clusters data set. Several measures showed different values over the number of clusters as compared to the clustering of the 1-cluster and n-cluster datasets. For instance, the coefficient J1 decreased in the interval c = 2 to c = 7, consistent with the fact that data can be clustered further (as seen for the n-cluster data); then it reached a plateau for higher number of classes, consistent with the fact that the data cannot be segregated any further (as the case of the 1-cluster data). The limit between the two behaviours was indeed at the true number of clusters (c = 7). This observation is valid for the other measures of compactness (e.g., FC, πm,1, IDintra) and separation (e.g., FS, IDinter, S, Km). When the data became noisy, some measures were less sensitive to the structure of the data (i.e., the presence of seven clusters). As illustrated in Fig. 2B, fuzzy separation FS and compactness πm,1 showed comparable behaviour as in the clustering of the n-cluster dataset, which reflects the influence of noisy distant points (low within-cluster compactness and between-cluster separation). Interestingly, in addition to Vdmin, quantities IDintra, Km and J1 were more robust to noise and showed high discriminability with an optimal value around the expected number of classes (Fig. 2B). This observation further motivated their inclusion in the definition of the new CV index.

Figure 2 Illustration of the behaviour of different measures of compactness and separation at different noise levels.

The behaviour of different measures of compactness and separation during FCM of the 7-cluster dataset with low (A, σ = 1) and high (B, σ = 4) noise levels. See full definition of the different measures in the ‘Methods’.

Figure 3 illustrates all CV indices for the 3-cluster, 7-cluster, and 11-cluster datasets with low noise level (σ = 1). All CV indices indicated the best c-partition for the expected number of clusters (maximum value for CVZLE, CVGV, CVPBM, CVWY, CVBWS, CVnew; minimum value for CVRLR and CVKP). Note that the new index CVnew is highly discriminative in pointing to the optimal c-partition. When the data became noisy (σ = 4), all CV indices, except CVBWS and CVnew, failed to indicate the optimal c-partition (Fig. 4). However, for data with higher copt (e.g., copt > 9), only the new index CVnew identified the true number of clusters, albeit with lower discriminability (e.g., compare Figs. 3B to 4B).

Figure 3 Plots of the CV indices for the simulated data at low noise level.

Plots of the CV indices for the simulated 3-cluster (A), 7-cluster (B) and 11-cluster (C) datasets with low noise level (σ = 1), when number of clusters increased from 2 to 19. All CV indices successfully indicated the expected number of clusters (copt = 3 in a, copt = 7 in b and copt = 11 in c). See full definition of these indices in the ‘Methods’.

Figure 4 Plots of the CV indices for the simulated data at high noise level.

Plots of the CV indices for the simulated 3-cluster (A), 7-cluster (B) and 11-cluster (C) datasets with high noise levels (σ = 4), when the number of clusters increased from 2 to 19. Only the new CV index identified the correct 11-partition at this level of noise.

An ad hoc analysis was conducted to monitor the behaviour of CVnew over different degrees of fuzziness m (m varying between 1.2 and 2.5), for a similar rationale see (Schwämmle & Jensen, 2010). This analysis showed that CVnew correctly identified the true number of clusters copt in almost all simulated datasets for m ∈ [1.2, 2.5], except for datasets with both high noise level (σ = 4) and high number of true clusters (copt > 9) where CVnew failed to identify copt when m ≥ 2 (i.e., CVnew underestimated copt at higher m values including the popular value of m =2). This ad hoc analysis confirmed the initial choice of m = 1.5.

FCM real fMRI data

As expected, the number of iterations for the convergence of the FCM algorithm varied across the 10 different initialisations. However, for a given c value and across the ten runs, the obtained c-partitions were very similar and the function Jm (Eq. (1)) reached the same minimum value (except for c values between 12 and 15 where one initialisation reached a different minimal Jm value compared to the other nine initialisations).

Identified clusters

Figure 5 plots the different coefficients and CV indices against the number of expected clusters c varying from 2 to 39. Measures such as IDinter, and Vdmin showed an interesting pattern when c increased, with high and decreasing values for small number of clusters (c < 10) and low and fixed values when c increased (a comparable behaviour was also seen for FC). This mirrored their behaviour during the clustering of the 1-cluster and n-cluster datasets. The change in the nature of the dependency occurred around c = 13, indicating the maximum c value that ensured different centroids V. For a number of expected clusters bigger than 13, the c-partition contained a few redundant classes (identical centroids V). However, for c < 13 clusters, although the obtained classes were compact (e.g., high FC values), the separation between clusters was not optimal (see for instance Vdmax, S, and Km). More specifically, the fuzzy separation measures S and Km showed optimal values for higher numbers of expected clusters at c larger than 17 clusters. At this range, the c-partition contained at least three similar centroids.

Figure 5 Illustration of the results using real fMRI data.

(A) Different measures of compactness and separation and (B) the different CV indices. The number of clusters varied from 2 and 39.

Figure 5B illustrates the dependency of different CV indices with c. Some CV indices (e.g., CVRLR and CVKP) showed optimal values for low c values (maximal fuzzy compactness), whereas other CV indices (e.g., CVZLE, CVBWS, CVGV, and CVPBM) showed optimal values at an intermediate number of expected clusters (i.e., maximal fuzzy separation). Interestingly, the new index CVnew went through different phases (i.e., different plateaus), depending on the weight of fuzzy separation and compactness (a change of behaviour visible at c = 15). The new index CVnew reached its maximum value at c = 24 clusters, ensuring a good compromise between separation and compactness of the c-partition of this real fMRI dataset.

The results of the morphological granulometry at U > 0.5 are illustrated in Fig. 6. As expected, the size of very large regions tend to decrease with the number of expected clusters, as large regions were subdivided further into smaller regions at higher c values. Interestingly, for each c-partition, the total number of single-voxel regions over all clusters was less than 0.04% of n (Fig. 6). Given the spatial smoothness of the fMRI data, there was no cluster containing exclusively single-voxel regions.

Figure 6 FCM results at different c values.

(A) Regions’ sizes (in number of voxels) for each crisp c-partition (at an arbitrary threshold of U > 0.5). Each dot (diamond shape) represents the size of one region in any cluster of the c-partition (c varying between 2 and 39). A base 10 logarithmic scale is used for the y-axis. (B) Total number of single-voxel regions for each c-partition. For the winning FCM partition (c = 24 with CVnew), there was less than 4 single-voxel regions per cluster on average. Total number of voxels n = 227,716; voxel size = eight mm3.

Figure 7 illustrates all obtained clusters for c-partitions with low fuzzy separation (c = 8), without redundant clusters (c = 13), at high fuzzy separation (c = 18), and at the optimal c value that maximised CVnew (c = 24). Identified voxels within the auditory cortex (i.e., voxels of interest) are shown in the first axial slice of each c-partition. Voxels in the auditory cortex were grouped with those in the occipital lobe at small c values (c = 8), but they became clearly segregated at larger c values (e.g., c = 18 and c = 24). Interestingly, identified voxels within the auditory cortex in the c-partition with 24 clusters were remarkably similar to those identified with model-based SPM methods (e.g., SPM map at p < 0.05 FWE-corrected, Fig. 8). Last but not least, the centroid of the relevant cluster with activations in auditory regions (Cluster “1” of the 24-partitions in Fig. 7) was strongly correlated with the experimental block design (r = 0.7, p < 0.001).

Figure 7 FCM results at different c values (A: c = 8, B: c = 13, C: c = 18 and D: c = 24).

Each obtained cluster (a 3D image) of each c-partition is illustrated by its most representative axial slice, with U values varying from 0.1 to 1.0. Cluster label is shown at the top-left corner of each axial slice (in white) and the MNI-z coordinate is indicated in black. For illustration purposes, the cluster that contained the expected activated voxels within the auditory cortex is labelled as Cluster ‘1’. The scatter plot (E) illustrates the correlations between the centroids of the 24-partition and the experimental block design (y-axis) against the fuzzy cardinality (cf. Eq. (6)) of each cluster (x-axis). Only one cluster showed significant correlation (p < 0.001) with the experimental design (r = 0.7). The fuzzy cardinality was divided by the total number of voxels, which would approximately reflect the ‘proportion’ of voxels contained in each cluster (average proportion around 4% (=1/ c)). Using the spatial location of the clustered voxels, one can potentially interpret the results of the FCM 24-partition (D). For example, Cluster 1 is showing auditory activations (cluster of interest) that highly correlated with the experimental block design (r = 0.7); Clusters 2–4 illustrate voxels in the visual system; Clusters 5–8 illustrate cerebellar and subcortical regions; Clusters 9–10 illustrate different medial parts of the default mode network; Clusters 11–12 contain voxels in ventral brain regions that are prone to MR signal loss; Clusters 13 and 14 are dominated by motion artefacts; Cluster 15 mainly shows CSF voxels; Clusters 19–24 contain white matter voxels. L, left hemisphere; R, right hemisphere.

Figure 8 SPM’s results.

SPM results illustrated with the function ‘montage’ of SPM12, with axial slices varying between MNI-z = −16 mm to MNI-z = +36 mm. (A) Results at a very liberal threshold of p < 0.05 uncorrected, (B) at p < 0.05 FWE-corrected. L, left hemisphere; R, right hemisphere.

Discussion

Using both simulated and real fMRI data, this study explored the usefulness of CV indices in identifying the best c-partition with FCM. This study also examined the behaviour of different compactness and separation measures, defined here as building blocks of the different CV indices. The optimal number of clusters varied with different CV indices, given that measures of compactness and separation were influenced by different features of the fMRI data (e.g., the expected high number of clusters, noise, and the amount of artefacts). A new CV index (CVnew) was introduced here and it showed relatively good robustness when clustering noisy data with high number of classes. Our study also highlighted the importance of analysing different measures of separation and compactness in order to get a better understating of the complex structure of the data.

The typical low signal-to-noise ratio in fMRI might be the most challenging issue that can hinder the success of clustering techniques. Here, simulated data were based on Gaussian-like noise distributions, and the success of different CV indices depended on the level of noise in the data. Our findings are in line with previous studies that compared several CV indices on different simulated datasets and found that CV indices may fail to indicate the true number of clusters in noisy data that have high number of classes (Suleman, 2017; Wang & Zhang, 2007; Zhou et al., 2014). It might be the case their effectiveness might even be lesser given the complex nature of noise in MRI images with significant correlations between voxels (Gudbjartsson & Patz, 1995; Parrish et al., 2000). To ensure better data input to FCM, it is thus recommended to use different pre-processing techniques that can reduce the impact of noise and improve data quality (Caballero-Gaudes & Reynolds, in press). The usefulness of such techniques with FCM on fMRI data warrants further studies.

Perhaps more importantly, the results stressed the importance of reading the behaviour of different separation and compactness measures, defined here as building blocks of CV indices, in order to depict an accurate description of the fMRI data (cf. Fig. 5). This is because it is most likely that there are different meaningful c-partitions depending on the scale at which the different clusters (i.e., brain networks) are segregated. Accordingly, it is not always useful to bias the analysis towards one elusive single c-partition, but rather appreciate that fMRI data might encompass different plausible patterns or networks at different spatio-temporal scales (Orban et al., 2015). Put another way, users need to relax the assumption that copt must be unique, and look instead for complementary explanations of the data at different copt values. For instance, using fuzzy clustering on resting-state fMRI data, Lee and colleagues (Lee et al., 2012) identified two optimal c-partitions with seven and eleven clusters that minimised a cluster dispersion measure (used as a CV index). Interestingly, the c-partition with 11 clusters further subdivided some of the clusters identified in the c-partition with seven clusters (Lee et al., 2012), most probably due to the known hierarchical organization of the brain networks. Our results of the clustering of real fMRI data also showed similar trends with clusters being further segregated with increasing number of expected clusters (e.g., compare clusters with c = 8 to clusters with c = 18 in Fig. 7).

Previous work suggested that, when CV indices fail to agree on the true number of clusters for high-dimensional datasets, a combination of different indices into a single index should be considered (Sheng et al., 2005; Zhou et al., 2014). Specifically, by using a weighted sum of several normalized CV indices, it has been shown that this weighted sum can improve the confidence of clustering solutions. Ultimately, this approach aims to force an agreement between CV indices so that one optimal single c-partition is selected. However, this approach may not be applicable to all contexts because: (i) the number and types of CV indices to be combined are arbitrary, (ii) there is no objective procedure to set optimal weights, and previous empirical work showed that such weights are data-dependent (Zhou et al., 2014), (iii) the weighted sum does not properly deal with redundant information, given that CV indices are likely to share similar compactness or separation measures, (iv) the relationships of some CV indices with the number of expected clusters can take any arbitrary shape (e.g., Fig. 5B), hence linear combinations may not be suitable, and (v) this approach implicitly assumes that there must be one unique ‘true’ explanation of the data. Here I argue that summation of different CV indices might not be useful for fMRI data clustering, because it ignores the possibility that different plausible explanations (different c-partitions) exist for the same data. Differences between CV indices should not be overlooked because they tend to highlight different existing features in the data.

The existence of different plausible explanations (c-partitions) of the same fMRI data can be further illustrated when examining the different compactness and separation measures used in the definition of the new CV index. More specifically, as illustrated in Fig. 5B, CVnew went through three different phases: (i) low values for c < 15, (ii) a plateau with high optimal values for 15 < c < 28, and (iii) another plateau for c > 28. The three phases indicated different segregated data structures depending on the predominance of either compactness or separation measures (Fig. 5A). For example, high fuzzy separation with well-isolated clusters was only achieved at c > 15, as reflected in the behaviour of Km and IDintra; however, when c increased the c-partitions became less compact (see FC), with higher fuzzy overlap and over-classification when c increased beyond 28 clusters (see IDinter). Given the expected small proportion of task-related activations in the auditory cortex, a segregation of relevant auditory voxels was only achieved with c > 15 clusters, for a similar rationale see (Chuang et al., 1999). In sum, looking at different compactness and separation measures, in addition to CVnew index, can provide a richer representation of the clustering results so that users can select the most useful c-partition among many potential possibilities.

Other methodological issues warrant further investigations. For instance, it might be interesting to test these CV indices with other varieties of FCM algorithms that incorporated spatial constraints during the minimisation of the objective function Jm (e.g., Ahmed et al., 2002; Liew, Leung & Lau, 2000), which can take into account the inherent spatial dependencies between neighbouring voxels (e.g., dependencies inflated by the spatial resampling and smoothing in fMRI). This would for instance penalise implausible solutions (c-partitions) with isolated voxels (e.g., Fig. 6). In addition, if outlier voxels existed in a dataset, this would artificially yield optimal CV values for c-partitions with a small number of clusters. In this context, it is useful to combine these CV indices with robust clustering techniques (for a review see Dave & Krishnapuram, 1997), adaptive distance measures (Tang et al., 2015), or other modified fuzzy clustering algorithms (e.g., Dik et al., 2014; Kao & Huang, 2013; Keller, 2000; Seghier, Friston & Price, 2007). Another challenging issue is to give meaning to the different identified clusters. Typically, users have to set objective criteria to distinguish relevant clusters from noise or artefact-driven clusters. For instance, for task-related fMRI data, clusters of interest are expected to have centroids similar (highly correlated) to the paradigm (Chuang et al., 1999; Fadili et al., 2000; Goutte et al., 1999; Jahanian, Soltanian-Zadeh & Hossein-Zadeh, 2005), as illustrated in Fig. 7. For task-free fMRI data, irrelevant clusters should be discarded, including clusters that are less consistent across sessions (Levin & Uftring, 2001) or when they include irrelevant brain voxels (e.g., in the white matter, ventricles, cerebrospinal fluid, arteries) (Ma et al., 2011).

Although FCM can provide useful data-driven explanations, deciding which clustering method is best for fMRI data remains an open question (Derntl & Plant, 2016). Typically, selecting a specific clustering algorithm entails a trade-off between different criteria (e.g., accuracy versus stability Thirion et al., 2014), with different methods may yield different clustering solutions. Many previous fMRI studies for instance have compared FCM against other data-driven methods, but findings varied considerably across studies, probably due to differences in fMRI data features in particular in terms of contrast-to-noise ratio and the level of physiological noise (Baumgartner et al., 2000; Dimitriadou et al., 2004; Lange et al., 2006; Wismuller et al., 2004). One popular data-driven method in the current literature is independent component analysis (ICA). ICA allows the detection of unexpected brain responses to stimuli, dissociation of functional networks and can be used as a powerful denoising tool (Stone, 2002). Previous work (Meyer-Baese, Wismueller & Lange, 2004; Smolders et al., 2007) have shown that FCM may outperform ICA when analyzing task-related fMRI data with good contrast-to-noise ratio. Nonetheless, it is fair to say that any comparison between ICA and FCM is an empirical question that is contingent on the nature of the fMRI data, the exact parametrization of FCM (Schwämmle & Jensen, 2010), the type of ICA algorithm, and the number of independent components (McKeown, Hansen & Sejnowsk, 2003).

Conclusions

Unsupervised FCM with different CV indices is a useful tool for analysing model-free fMRI datasets, an alternative to the widely used independent component analysis methods. It is recommended to combine different CV indices in order to draw a complete picture of the structure of the data. The assumption here is that different CV indices may point to different optimal c-partitions, given the heterogeneous behaviour of many measures of compactness and separation. Rather than discarding discrepancies between CV indices, such discrepancies should be appreciated because they reflect the hierarchical organization of brain networks. This was clearly visible for instance when analysing the different phases of the plot of the new CV index against the number of clusters. Overall, the existence of different c-partitions for the same fMRI data should not be overlooked in future clustering studies.

Supplemental Information

Supplemental Information 1 Supplementary Figures 1 to 3

Click here for additional data file.

Supplemental Information 2 Code (Matlab) used to generate 22 different simulated datasets

Click here for additional data file.

Additional Information and Declarations

Competing Interests

Author Contributions

Data Availability

The authors declare there are no competing interests.

Mohamed L. Seghier conceived and designed the experiments, analyzed the data, contributed reagents/materials/analysis tools, prepared figures and/or tables, authored or reviewed drafts of the paper, approved the final draft.

The following information was supplied regarding data availability:

The fMRI dataset used in this paper is freely available on SPM’s website (http://www.fil.ion.ucl.ac.uk/spm/data/auditory.html) as explicitly mentioned in the ‘Methods’. A code that describes how artificial data were generated and the core steps of the classic FCM algorithm is provided as Supplemental File.

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
