# Peer review of "Clustering of fMRI data: the elusive optimal number of clusters"

_PeerJ, doi:10.7717/peerj.5416_

## Round 0.1 · original submission · Minor Revisions

Dear Authors,

Please perform the relevant revisions to the manuscript as suggested by the two peer reviewers.

·

Basic reporting

Grammar, style, typo:
- line101: “the” is missing in front of “same”
- In Results section and later, it is often not clear whether the word "cluster" refers to the data (ground truth) or the method (expected). E.g. in lines 434 and 507, I guess the latter. I recommend adding the word "expected" whenever appropriate.
- line 407: I think ID_inter should be ID_intra. Because in Figure 2B, it is ID_intra and not ID_inter, which is discriminative. Also, V_dmin seems also quite robust (i.e. discriminative at both noise levels).

Field background:
- The authors provide a very concise description of the fuzzy clustering.

Literature references:
- I find it strange that no CV indices later than 2006 is considered despite developments (e.g. (DOIs): 10.1016/j.fss.2010.07.005, 10.1109/ICSSEM.2011.6081293, 10.1016/j.fss.2013.12.013, 10.1007/s00500-016-2453-y); and that there is no reference to a recent comparison study (DOI: 10.15837/ijccc.2014.3.237).

Figures:
- Figure 6: The layout of figure 6 is very confusing an uncommon in neuroimaging literature. I recommend presenting sequential slices at the same position for all c values, as well as the SPM results. Slice locations (i.e. z-coord in MNI space) and laterality (R-L or L-R) should be also marked.

Experimental design

Research approach:
- It is overall well designed: I would like to emphasize the rationale of using a modified version of D (Eq. 5) and the inclusion criteria for CVs (lines 213-217).
- The interaction between fuzziness (m) and CV indices is very important; however, I found the theoretical discussion of larger m values a bit pointless, when the authors decided to stay within the limit based on the literature. I would (also) rather like to see some analysis on how robust the CV indicies within the range of m = 1.2-2.5.

Methodological issues:
- Reslicing a data to resolution higher than the acquisition 2x2x2mm3 vs 3x3x3mm3 is unfortunately a common mistake. It unnecessarily increases the number of voxels without adding new information. It should be particularly avoided when high dimensionality and voxel redundancy are the issues.

Validity of the findings

Speculation:
- line 523: Suggestion of spatial constraints seems to miss the fact that, although, it is true that neighboring voxels are dependent from each other; voxels far from each other might also belong to the same (perhaps not congruent) cluster, due to the functional connectivity.

Conclusion
- line 543: Combining different CV indices is one of the most important suggestion of the manuscript! Yet, the largest part rather focuses on the (well-supported) superiority of the new CV_new, and the integrative aspects is only demonstrated by looking at the effect of c value on CV_new.
a. Data is robust, statistically sound, & controlled.
b. Conclusion are well stated, linked to original research question & limited to supporting results.

·

Basic reporting

The paper is well written, the objectives are clear, and method/results presented appropriately. While the code for simulated data is available, there is no code to check the actual clustering, which should also be provided.

Experimental design

The overall approach is comprehensive testing many of the ‘popular’ CV along with the new one, I have no concern here.

Validity of the findings

My only issue is with testing of the fMRI data and the lack of consideration for the smooth spatial nature of the data (acknowledged in the discussion). Without proposing here to incorporate this into the clustering, I suggest 2 additional (small) analyses:
A) Add a measure of spatial ‘isolation’ ; for instance results from the MoA data suggest maybe 3 or 4 scales / clustering levels - could you plot along with the clustering number the median number of isolated voxels and median cluster size in space? Say 8 clusters is what you choose as a partition, what are there spatial size and how many isolated voxels – we know isolated voxels don’t make much sense
B) Given the data driven approach, why only comparing to the MoA auditory activation map? B.1) I suggest again for the 3/4 scales / clustering levels to correlate each cluster time course with the design to get an idea of which one follows the experimental time course B.2) check/report better on spatial structures, depending the clustering solution, we can easily interpret some cluster as motion or csf; B.3) running GIFT with infomax I got 35 components and only one fitting the design; how does that compare?. I understand the goal is not to compare to ICA, but ICA is quite popular and clustering is not despite the advantage of possibly showing hierarchical levels of segregations which is a strong ‘selling’ point, hence maybe worth comparing the maps (again say the 3 levels of clustering vs ICA)

---

## Round 0.2 · accepted · Accept

Dear Authors,Congratulations,the revised manuscript has been accepted to be published in PeerJ. Thanks.

# ·

Basic reporting

no comment

Experimental design

no comment

Validity of the findings

no comment

Additional comments

I welcome the improvement of the manuscript and accept the authors' answers when appropriate.

·

Basic reporting

no comment

Experimental design

Now that the additional code is provided, this makes things more transparent.

Validity of the findings

no comment